# Identification of a Partial and Selective TRPV1 Agonist CPIPC for Alleviation of Inflammatory Pain

**DOI:** 10.3390/molecules27175428

**Published:** 2022-08-25

**Authors:** Liying Dong, Qiqi Zhou, Qianqian Liang, Zhen Qiao, Yani Liu, Liming Shao, Kewei Wang

**Affiliations:** 1Departments of Pharmacology, School of Pharmacy, Qingdao University Medical College, Qingdao 266073, China; 2Department of Pharmacology, Qilu Medical University, Zibo 255300, China; 3Department of Medicinal Chemistry, School of Pharmacy, Fudan University, No. 826 Zhangheng Road, Shanghai 201203, China; 4Institute of Innovative Drug Discovery, Qingdao University Medical College, 38 Dengzhou Road, Qingdao 266021, China

**Keywords:** TRPV, desensitization, whole-cell patch clamp, site-directed mutagenesis, dorsal root ganglion and antinociception

## Abstract

Transient receptor potential vanilloid 1 (TRPV1) is a non-selective cation channel, predominantly expressed in a subset of peripheral sensory neurons for pain signaling. Topical application of agonist capsaicin for desensitizing TRPV1 currents has been approved for relief of chronic pain. However, the potent TRPV1 capsaicin is not ingestible and even topical capsaicin causes common side effects such as skin irritation, swelling, erythema and pruritus, suggesting that a mild TRPV1 agonist might be helpful for reducing side effects while reliving pain. In this study, we reported on a partial and selective TRPV1 agonist 4-(5-chloropyridin-2-yl)-*N*-(1*H*-indazol-6-yl)piperazine-1-carboxamide named **CPIPC** that was modified based on targeting the residue Arg557, important for conversion between the channel antagonism and agonism. Whole-cell patch clamp recordings indicated a concentration-dependent activation of TRPV1 currents by **CPIPC** with an EC_50_ of 1.56 ± 0.13 μM. The maximum efficacy of **CPIPC** (30 μM) was about 60% of saturated capsaicin (10 μM). Repetitive additions of **CPIPC** caused TRPV1 current desensitization in both TRPV1-expressing HEK293 cells and dorsal root ganglion (DRG) sensory neurons. Oral administration of **CPIPC** dose-dependently alleviated inflammatory pain in mice. Further site-directed mutagenesis combined with molecular docking revealed that residue Arg557 is critical for TRPV1 activation by **CPIPC**. Taken together, we identified a novel partial and selective TRPV1 agonist **CPIPC** that exhibits antinociceptive activity in mice.

## 1. Introduction

Transient receptor potential vanilloid 1 (TRPV1) is a well-known nonselective ligand-gated cation channel that is activated by noxious heat (>42 °C) and mainly expressed in peripheral sensory neurons, such as trigeminal and dorsal root ganglia [1,2]. TRPV1 plays an important role in the inflammatory pain induced by tissue injury or in the development of inflammatory pain [2,3]. Inflammatory mediators (ATP, bradykinin and prostaglandins) can sensitize TRPV1 through reducing temperature threshold of channel activation [4,5,6,7,8,9,10]. TRPV1 sensitization leads to the nociceptor hyperexcitability during tissue inflammation [11], and desensitizing TRPV1 by suppressing channel function can relieve pain [12,13,14].

Preclinical studies or clinical trials indicate that genetic or pharmacological inhibition of TRPV1 can cause hyperthermia through the mechanism elevating heat sensing threshold [15,16,17]. In contrast, a potent TRPV1 agonist capsaicin (Figure 1) by topical application has been widely used for clinical therapy of pain [18,19]. However, topical application of capsaicin can induce adverse effects including hypothermia, pruritus, pain, erythema, and papules at the administration site [20]. Recently, a partial TRPV1 agonist MDR-652 (Figure 1) shows antinociceptive activity in the mouse models of inflammatory and neuropathic pain but with less toxicity compared with capsaicin, suggesting that development of partial TRPV1 agonists with subpotency might be an effective strategy for pain relief [21].

The cryo-EM structural investigations illustrate that both TRPV1 agonists (capsaicin and RTX) and antagonists (capsazepine) can fit into the same binding pocket consisting of residues such as Tyr511, Met514, Leu518, Leu547, Thr550, Arg557, Glu570, and Leu670 [22]. The structural complexes of TRPV1 with vanilloid ligands reveal that the hydrogen bonds with residue Arg557 are essential for the channel activation by agonist capsaicin or RTX, but not antagonists. This suggests that the Arg557 might be critical for a switch between agonism and antagonism [23]. Therefore, we hypothesized that modification of TRPV1 antagonists for forming a hydrogen bond with Arg557 might lead to identification of partial TRPV1 agonists.

To test this hypothesis, we synthesized and identified a lead compound 4-(5-chloropyridin-2-yl)-*N*-(1*H*-indazol-6-yl)piperazine-1-carboxamide (**CPIPC**) based on hybridization of BCTC and ABT-102 (Figure 1). BCTC, a representative TRPV1 antagonist with strong affinity, is only a tool molecule used for pharmacological studies because of its poor aqueous solubility, short half-life and limited metabolic stability. ABT-102, 1,3-disubstituted urea-derivative with drug-like properties is another potent and selective TRPV1 antagonist that displays analgesic efficacy in pain models of inflammation, post-operative, osteoarthritic and bone cancer pain [24]. Electrophysiological recordings revealed that **CPIPC** was a partial and selective TRPV1 agonist, with an EC_50_ value of 1.56 ± 0.13 μM and an efficacy about 61.82 ± 0.49% of capsaicin-mediated channel current saturation. **CPIPC** was also able to desensitize TRPV1 currents in DRG sensory neurons. Oral **CPIPC** showed antinociceptive effects in the mouse model of inflammatory pain. Mechanistically, site-directed mutagenesis combined with molecular docking further confirmed that Arg557 is critical for TRPV1 activation by **CPIPC**.

## 2. Results

### 2.1. Concentration-Dependent Activation of TRPV1 by the Selective and Partial Agonist CPIPC in TRPV1-Transfected HEK293 Cells

We started testing the effect of **CPIPC** on *h*TRPV1 transiently transfected HEK293 cells in calcium-imaging assay. As shown in Figure 2A, application of 1 μM **CPIPC** or 1 μM capsaicin as a positive control induced a robust Ca^2+^ influx in TRPV1 transfected HEK293 cells compared with blank HEK293 cells. Ionomycin, as a positive control of calcium ionophore, resulted in a drastic elevation of intracellular calcium in both TRPV1-overexpressing and non-transfected HEK293 cells. In the FlexStation 3 multi-mode microplate reader assay, **CPIPC** at 0.1 μM or 10 μM also increased intracellular levels of calcium in TRPV1-transfected HEK293 cells, which was completely reversed by a selective TRPV1 antagonist BCTC at 10 μM (Figure 2B). These results suggest that **CPIPC** is an agonist of TRPV1.

To further confirm the activation of TRPV1 by **CPIPC**, we performed the whole-cell patch clamp recordings on TRPV1 transiently transfected HEK293 cells. As shown in Figure 3A,B, **CPIPC** at 1 μM activated a large outward rectification TRPV1 current, and the effect could be wash-out before further activation of TRPV1 by capsaicin (1 μM). Perfusing different concentrations of **CPIPC** from 0.1 μM to 30 μM resulted in a concentration-dependent activation of TRPV1 currents with EC_50_ of 1.56 ± 0.13 μM and a Hill coefficient of 2.07 ± 0.20 (Figure 3C,D). The maximum efficacy induced by **CPIPC** (30 μM) was 61.82 ± 0.49% of the current induced by saturated capsaicin (10 μM) (Appendix A). We also observed a further increase of saturated 30 μM **CPIPC**-induced currents after adding a mixture of saturated capsaicin (10 μM) and **CPIPC** (30 μM) (Figure 3E), indicating **CPIPC** was a partial TRPV1 agonist. The **CPIPC**-mediated TRPV1 currents were concentration-dependently inhibited by a selective TRPV1 antagonist capsazepine ranging from 0.1 to 10 μM (Figure 3F). Repetitive applications of **CPIPC** (1 μM) caused a progressive reduction of TRPV1 currents by 79.33 ± 0.79% (Figure 3G,H), indicating a TRPV1 desensitization induced by **CPIPC**. The **CPIPC**-induced desensitization also reduced the potency of capsaicin about 10-fold (Appendix A), demonstrating the reduced TRPV1 activation by capsaicin under **CPIPC**-induced desensitization.

To evaluate the selectivity of **CPIPC**, we tested the effect of **CPIPC** on *m*TRPV2, *h*TRPV3, *h*TRPV4, *h*TRPA1 and *h*Nav1.7 channels using a whole-cell patch-clamp assay. As shown in Figure 4A, 1 μM **CPIPC** had no detectable activation of TRPV2 channel that was activated by agonist 2-APB [25]. Similarly, **CPIPC** at 1 μM caused no activation of TRPV3, TRPV4 or TRPA1 currents (Figure 4B–D), as compared with the channel activation by their agonists 2-APB for TRPV3, GSK1016790A for TRPV4 or AITC for TRPA1 [25,26,27]. We also evaluated the effect of **CPIPC** or capsaicin on Nav1.7 channels expressed in HEK293 cells. As shown in Appendix A, 1 μM capsaicin induced an inhibition of Nav1.7 current about 19.14 ± 0.35%, whereas 1 μM **CPIPC** resulted in reduction of Nav1.7 current about 4.01 ± 0.55%. These results demonstrated that **CPIPC** was a selective TRPV1 agonist.

### 2.2. TRPV1 Desensitization Induced by Repetitive Applications of CPIPC in DRG Neurons

We next tested the effect of **CPIPC** on native TRPV1 currents recorded from dissociated mouse DRG neurons at +80 mV. In whole-cell patch-clamp recordings, we observed that 10 of 19 (53%) small-or medium-diameter DRG neurons were sensitive to both capsaicin (10 μM) and **CPIPC** (10 μM) with an amplitude of 1.37 ± 0.22 nA to **CPIPC** (*n* = 6; range of 0.97–2.18 nA). As shown in Figure 5A, **CPIPC** at 10 μM elicited a large capsaicin-sensitive outward currents in DRG neuron at depolarizing potential at +80 mV, and repetitive 5 times administrations of **CPIPC** led to a desensitization with 67.72 ± 1.08% reduction (Figure 5B). These results demonstrate that **CPIPC** can activate the native TRPV1 currents and also cause the current desensitization upon repetitive applications in capsaicin-sensitive nociceptive DRG neurons.

### 2.3. Pungency Study of CPIPC in Mice

It has been reported that topical application of TRPV1 agonists like capsaicin could induce pain-producing effects in the mouse [28,29]. Therefore, we evaluated the pungency of **CPIPC** in mouse eye-wiping test. As a positive control, instillation of 10 μL capsaicin (10 μg/mL) solution into mouse left eyes induced the wiping times of 15.67 ± 0.84 within 60 s (Figure 6A). In contrast, instillation of **CPIPC** solution (10, 30, 100 μg/mL) only evoked 4.33 ± 0.61, 5.50 ± 0.56 and 5.67 ± 0.76 wiping movements, respectively.

### 2.4. Antinociceptive Effects of CPIPC on Mouse Model of Inflammatory and Thermal Pain

We tested the antinociceptive effect of **CPIPC** on formalin-induced pain model with intraplantar injection of 5% formalin into mouse hind paws. As a positive control, pre-oral administration of a nonsteroid anti-inflammatory drug ibuprofen (2 mg/mL) for 60 min significantly reduced the time of hind paw licking induced by formalin in Phase II but not in Phase I, as compared with the vehicle control (Figure 6B,C). In contrast, pre-oral administration of **CPIPC** at 10 mg/kg significantly relieved pain behavior induced by formalin in Phase I. 3 and 10 mg/kg **CPIPC** caused a dose-dependent reduction of formalin-induced licking in Phase II. These observations demonstrated an antinociceptive effect of **CPIPC** on inflammatory pain.

We further tested the anti-nociceptive effect of **CPIPC** on inflammatory pain induced by intraplantar injection of CFA into mouse hind paws. The paw withdrawal mechanical threshold (PWMT) was measured by von Frey hair assay after injection of CFA for 24 h. As shown in Figure 6D and Table 1, the injection of CFA induced a mechanical allodynia as compared to the control group. In contrast, oral administration of **CPIPC** (1, 3, 10 mg/kg) dose-dependently increased the PWMT and the effect lasted for about 2 h. As a positive control, ibuprofen (2 mg/kg, *p.o.*) also reduced CFA-induced inflammatory pain, similar to the anti-nociceptive activity observed in the effect of **CPIPC** at 3 mg/kg.

### 2.5. Identification of Residue Arg557 Critical for TRPV1 Activation by CPIPC

We adopted the structure of TRPV1 in a complex with RTX (PDB:5IRX) for molecular docking of **CPIPC** using the Gold 5.3.0 software (University of Cambridge, Cambridge, UK). Our docking shows that **CPIPC** is inserted into a binding pocket formed by residues between S4 and the S4–S5 linker and the indazole of **CPIPC** forms two hydrogen bonds with Arg557 and Ala566 (Figure 7A,B), similar to capsaicin, that also forms a hydrogen bond with the Arg557 in the same binding pocket (Figure 7C).

To test whether the region of **CPIPC** predicted to contact Arg557 is critical for binding, we made chemical modifications to **CPIPC** by substituting the indazole ring with indole (**CPIPC-1**) or methyl (**CPIPC-2**) groups for disrupting the formation of the hydrogen bond with Arg557 and Ala566, respectively (Figure 7D). When **CPIPC-1** had its indazole ring replaced by indole group it resulted in a decrease of TRPV1 activation. In contrast, for **CPIPC-2**, the –NH– of indazole was substituted by a methyl group while remaining the hydrogen bond with Arg557 had a stronger activation on TRPV1 (Figure 7E,F), as compared to **CPIPC-1**. This indicates that the region of CPIPC predicted to contact Arg557 is critical for activating TRPV1 by **CPIPC**.

We further constructed a mutant R557A and tested the effect of **CPIPC** on TRPV1 activation. Mutating Arg557 caused a dramatic attenuation of **CPIPC**- (or capsaicin-) mediated TRPV1 activation (Figure 7G and Appendix A), consistent with the observation that residue Arg557 is critical for capsaicin-evoked response [30].

## 3. Discussion

Desensitizing TRPV1 is known to induce a long-term defunctionalization of nociceptors for pain relief, serving as an attractive strategy to develop new antinociceptive agents. Over the past years, several TRPV1 agonists have been reported for their antinociceptive activities, including capsaicin, Muralatin L and MDR-652 [21,31,32]. However, potent TRPV1 agonist capsaicin causes an initial adverse effect of pungency and even an 8% capsaicin dermal patch can lead to erythema and pain at the site of application [33]. In contrast, a partial TRPV1 agonist MDR-652 and a subpotent capsaicin analogue YB-16 have been shown to be much weaker in causing pungency [21,34]. These observations suggest developing subpotent TRPV1 partial agonists might be an effective strategy to treat inflammatory and neuropathic pain with less toxicity.

Our observations show that **CPIPC** can dose-dependently relieve formalin- and CFA-induced nocifensive behaviors in male mice. Injection of formalin into mouse hind paws induces pain responses in two phases: phase I (Figure 6B), an acute pain stage caused by direct stimulation of C-fiber nociceptors; and phase II (Figure 6C), the later phase of inflammatory response [32]. Complete Freund’s Adjuvant (CFA) is a widely used reagent for generation of rodent models of inflammation. The injection of CFA into the hind paws of mice or rats results in both mechanical allodynia and thermal hyperalgesia within 24 h. The inflammation condition occurs for 5 days before a slow recovery [35,36]. **CPIPC** (1, 3, 10 mg/kg) dose-dependently alleviated the CFA-induced mechanical allodynia and the effect lasted for about 2 h in mice. **CPIPC** as a partial TRPV1 agonist is different than other TRPV1 agonists, such as capsaicin which causes burning sensation and is only for topical use. **CPIPC** can be internally used, holding developmental potential for an antinociceptive agent.

The cryo-EM structures of TRPV1 in complex with capsaicin and capsazepine have provided a framework for predicting agonist binding sites in TRPV1 [23,37]. TRPV1 vanilloid agonists occupy and displace the resident phosphatidylinositol lipid in the binding pocket between S4 and S4–S5 linker. The vanillyl rings of these agonists, such as RTX, function to stabilize the interaction between Arg557 and Glu570 for facilitating the movement of the S4–S5 linker away from the central axis of the channel, thereby opening the lower gate through coupled movements [23]. Such a mechanism is further supported by the mutations in S4/S4–5 regions where R557A, R557K and E570R slow TRPV1 activation rate via capsaicin [30]. Interestingly, the TRPV1 antagonist capsazepine, for example, also can fit into the binding pocket of agonists without its interaction with Arg557. Therefore, Arg557 is a key residue that TRPV1 agonists act on [23,38]. Targeting the Arg557 may lead to identification of TRPV1 agonists of different potency and efficacy.

In this study, our molecular docking reveals that **CPIPC** and capsaicin share a similar pose in the binding pocket formed by S4 and S4–S5 linker. Either indazole of **CPIPC** or phenolic hydroxyl group of capsaicin can form a hydrogen bond with Arg557, whereas mutating R557 alters **CPIPC**- or capsaicin-induced channel sensitivity. Besides R557, residues Y511 and S512 from S3 segment, and M547 and T550 from S4 segment, are also essential for agonistic effect of capsaicin as mutations Y511A or S512Y of rTRPV1 abolish the channel sensitivity to capsaicin. This indicates that these key residues also contribute to **CPIPC**-mediated activation of TRPV1 [39,40,41].

In general, our identification of **CPIPC** not only provides a tool molecule but also a lead with therapeutic potential for pain therapy.

## 4. Materials and Methods

### 4.1. Reagents and Chemicals

The synthetic routes of 4-(5-chloropyridin-2-yl)-*N*-(1*H*-indazol-6-yl)piperazine-1-carboxamide (**CPIPC**), 4-(5-chloropyridin-2-yl)-*N*-(1*H*-indol-6-yl)piperazine-1-carboxamide (**CPIPC-1**) and 4-(5-chloropyridin-2-yl)-*N*-(1-methyl-1*H*-indazol-6-yl)piperazine-1-carboxamide (**CPIPC-2**) were shown as Appendix A. The NMR, HRMS and HPLC spectra of these compounds were shown in Appendix A. Capsaicin, 2-APB and carvacrol were purchased from Sigma-Aldrich (St. Louis, MO, USA). Compounds were dissolved in DMSO for stock solutions before further dilutions with cell bath solutions.

### 4.2. Synthesis of CPIPC Compounds

4-(5-chloropyridin-2-yl)-*N*-(1*H*-indazol-6-yl)piperazine-1-carboxamide (**CPIPC**). Yield, 92%. White solid. ^1^H NMR (400 MHz, DMSO-*d*_6_, ppm) δ 12.77 (s, 1H), 8.71 (s, 1H), 8.12 (s, 1H), 7.89 (s, 1H), 7.84 (s, 1H), 7.61 (d, *J* = 8.8 Hz, 1H), 7.56 (d, *J* = 8.3 Hz, 1H), 7.14 (d, *J* = 8.5 Hz, 1H), 6.91 (d, *J* = 9.1 Hz, 1H), 3.55 (d, *J* = 7.0 Hz, 8H). ^13^C NMR (151 MHz, DMSO-*d*_6_, ppm) δ 157.23, 154.97, 145.39, 140.42, 138.59, 137.07, 132.97, 119.80, 118.85, 118.25, 114.87, 108.45, 98.51, 44.39, 43.17. HR-MS (ESI^+^), calcd for C_17_H_17_ClN_6_O, [M + H]^+^
*m*/*z*: 357.1225, found: 357.1221.

4-(5-chloropyridin-2-yl)-*N*-(1*H*-indol-6-yl)piperazine-1-carboxamide (**CPIPC-1**). Yield, 73%. Light white solid. ^1^H NMR (400 MHz, DMSO-*d*_6_, ppm) δ 10.90 (s, 1H), 8.48 (s, 1H), 8.15 (d, *J* = 2.4 Hz, 1H), 7.68 (s, 1H), 7.63 (dd, *J* = 9.1, 2.5 Hz, 1H), 7.38 (d, *J* = 8.5 Hz, 1H), 7.23–7.18 (m, 1H), 7.03 (d, *J* = 8.4 Hz, 1H), 6.93 (d, *J* = 9.1 Hz, 1H), 6.32 (s, 1H), 3.56 (d, *J* = 7.1 Hz, 8H). ^13^C NMR (151 MHz, DMSO-*d*_6_, ppm) δ 157.28, 155.31, 145.39, 137.05, 135.90, 134.23, 124.13, 123.01, 119.12, 118.82, 113.44, 108.44, 102.66, 100.58, 44.43, 43.16.

4-(5-chloropyridin-2-yl)-*N*-(1-methyl-1*H*-indazol-6-yl)piperazine-1-carboxamide (**CPIPC-2**). Yield, 82%. Pink solid. ^1^H NMR (400 MHz, DMSO-*d*_6_, ppm) δ 8.80 (s, 1H), 8.12 (s, 1H), 7.87 (s, 1H), 7.81 (s, 1H), 7.61 (dd, *J* = 9.1, 1.0 Hz, 1H), 7.57 (d, *J* = 8.7 Hz, 1H), 7.16 (d, *J* = 8.7 Hz, 1H), 6.91 (d, *J* = 9.1 Hz, 1H), 3.91 (s, 4H), 3.55 (d, *J* = 8.9 Hz, 8H). ^13^C NMR (151 MHz, DMSO-*d*_6_, ppm) δ 157.22, 154.95, 145.38, 139.95, 138.79, 137.08, 131.84, 120.23, 118.97, 118.88, 115.03, 108.46, 97.99, 44.39, 43.16, 34.92.

### 4.3. Transient Transfection of TRPs and Cell Culture

The HEK293 cell line (1101HUM-PUMC000010) was obtained from the Cell Resource Center, Peking Union Medical College (which are the headquarters of National Infrastructure of Cell Line Resource, NSTI, Beijing, China). HEK293 cells were cultured in Dulbecco’s modified Eagle’s medium (DMEM) with 10% fetal bovine serum (FBS, VivaCell, Shanghai, China) at 37 °C with 5% of CO_2_. For whole-cell patch-clamp recordings, HEK293 cells placed on glass coverslips were transiently transfected with a mixture of 2 μL Lipofectamine 2000 (Invitrogen, Carlsbad, CA, USA), 2.5 μg human TRPV1 (NM_080704.3), mouse TRPV2 (NM_011706.2), human TRPV3 (NM_145068.4), human TRPV4 (NM_021625.5) and human TRPA1 (NM_007332.3) cDNA and 250 ng GFP in 100 μL Opti-MEM medium (Gibco^TM^, ThermoFisher, Grand Island, NY, USA) for 4 h [32]. Transfected cells were identified by expression of GFP as an indicator. For the FlexStation 3 multi-mode microplate reader assay and calcium-imaging assay, HEK293 cells were transiently transfected with a similar mixture but without GFP. After transfection for about 24 h, patch-clamp, the FlexStation 3 multi-mode microplate reader assay and calcium-imaging assay were performed. All cDNA clones were verified by sequencing.

### 4.4. Calcium-Imaging

TRPV1 transiently transfected or blank HEK293 cells were seeded on 1 cm^2^ glass coverslips for 24 h before being loaded with 4 μM Fluo-8 (ab142773, Abcam, Cambridge, MA, USA) in Ringer’s solution (pH 7.2) containing 140 mM NaCl, 5 mM KCl, 2 mM MgCl_2_, 10 mM d-glucose, 10 mM HEPES, and 2 mM CaCl_2_ for 30 min in 5% CO_2_ at 37 °C. The tested compounds were prepared in Ringer’s solution. After being washed twice with Ringer’s solution, cells were subsequently treated with **CPIPC** (1 μM), capsaicin (1 μM), and ionomycin (10 μM). Fluorescent signals were collected by a CCD camera (DS-Qi2) of Nikon powered by CellSens Dimension at 3-s intervals with a 20× objective lens. All experiments were performed at room temperature, and data were expressed as the mean ± SEM.

### 4.5. Measurement of Intracellular Calcium in FlexStation 3 Multi-Mode Microplate Reader Assay

The intracellular calcium level of HEK293 cells expressing TRPV1 channels was measured in the presence or absence of agonists or antagonists using the Screen Quest™ Calbryte-520 Probenecid-Free and Wash-Free Calcium Assay Kit (36318, AAT Bioquest, Sunnyvale, CA, USA) in FlexStation 3 Microplate Reader (Molecular Devices, San Francisco, CA, USA). TRPV1 transiently transfected HEK293 cells were seeded into a black 96-well plate with a density of 30,000 cells/well before cultured in 5% CO_2_ at 37 °C overnight. The calbryte™ 520 AM dye-loading solution was prepared by mixing 9 mL HBSS (137 mM NaCl, 5.5 mM d-glucose, 0.8 mM MgSO_4_, 0.4 mM KH_2_PO_4_, 0.1 mM Na_2_HPO_4_, 5.4 mM KCl, 1.3 mM CaCl_2_, 4 mM NaHCO_3_, and 20 mM HEPES, pH 7.4), 1 mL 10× Pluronic^®^ F127 Plus and 20 µL Calbryte™ 520 AM stock solution together. Cells were incubated with 100 µL/well Calbryte™ 520 AM dye-loading solution in 5% CO_2_ at 37 °C for 60 min and then immediately placed on a FlexStation 3 multi-mode microplate reader. Different concentrations of compounds were added onto cells at 17 s and corresponding TRP agonists (capsaicin, 2-APB and carvacrol) were added at 100 s. The whole test lasted for about 180 s. Fluorescence intensity was monitored at Ex/Em = 490/525 nm and measured at an interval of 1.6 s [12,25,32].

### 4.6. Disassociation and Culture of Mouse DRG Neurons

Dorsal root ganglion (DRG) neurons were obtained from 6–8-week-old C57BL/6 mice (Beijing Vital River Laboratory Animal Technology Co., Ltd., Beijing, China) as previously described [32,42]. The ganglia were digested with collagenase type 2 (1 mg/mL, Worthington, Lakewood, NJ, USA) and neutral protease dispase II (7.5 mg/mL, Merck, Kenilworth, NJ, USA) at 37 °C with 5% of CO_2_ for 30 min. DRG cells were mechanically dissociated and washed with DMEM plus 10% fetal bovine serum. After centrifugation at 1500 rpm, the DRG cells were suspended and plated on poly-d-lysine-coated glass coverslips. Electrophysiological recordings were carried out within 48 h.

### 4.7. Electrophysiology

Whole-cell patch-clamp recordings were performed on HEK293 cells co-expressing TRPV1-4 and GFP under fluorescent light at room temperature (23–25 °C) using a HEKA EPC10 amplifier with PatchMaster software (HEKA Harvard, Holliston, Church Hill, TN, USA). Currents were filtered at 2.9 kHz and sampled at 20 kHz. Borosilicate glass pipettes (BF150-86-100, Sutter Instruments, Novato, CA, USA) were pulled by the horizontal micropipette puller (P-97, Sutter Instruments, Novato, CA, USA) and fire polished by the MF-830 polisher (Narishige, Tokyo, Japan). The electrode resistance was 2~4 MΩ after being filled with pipette solution and liquid junction potential was adjusted to zero after the pipette-in-bath solution. For ramp recordings of HEK293 cells, both bath solution and pipette solution components were (mM): NaCl 130; EDTA 0.2; HEPES 3 [43]. Cells were held at 0 mV after use of whole-cell recording mode before given a ramp stimulus from −100 mV to +100 mV every 1 s. Series resistances (6~15 MΩ) were compensated by 60–80%. Gravity perfusion was administrated and the perfusion rate was adjusted to 0.5 mL/min. It is necessary to divide the whole-cell current (*I*_m_) by the cell capacitance (*C*_m_) to calculate the current density (*I*_d_).

TRPV1 currents from small- or medium-diameter (15–25 μm) DRG neurons were recorded using whole-cell voltage-clamp technique. The bath solution contains (mM): NaCl 137; MgCl_2_ 1; KCl 4; Glucose 10; HEPES 10 (pH 7.4 adjusted with NaOH, osmotic pressure 300 mOsm adjusted with sucrose). The pipette solution components were (mM): KCl 65; KF 75; MgCl_2_ 2; EGTA 5; HEPES 10 (pH 7.2 adjust with KOH, osmotic pressure 300 mOsm-adjust with sucrose). After the whole-cell recording mode is formed, cells were given a 300-ms step stimulus to +80 mV, followed by a 300-ms step stimulus to −80 mV, at 1-s intervals with a holing potential at 0 mV.

### 4.8. Animals

Adult male C57BL/6 mice (7 weeks) were purchased from Beijing Vital River Laboratory Animal Technology Co., Ltd., (Beijing, China). All experimental procedures were approved by the Animal Ethics Committee of Qilu Medical University and complied with the ethical guidelines of the International Association for the Study of Pain. Animals were housed in a 12 h alternating light/dark cycle room with free access to food and water.

### 4.9. Complete Freund’s Adjuvant (CFA)–Induced Inflammatory Pain Model

Adult male C57BL/6 mice were anesthetized with isoflurane and then intraplantarally injected 20 µL complete Freund’s adjuvant (CFA) (Sigma, St. Louis, MO, USA) into the right hind paw. In the control group, 20 µL saline was injected into the right hind paw of mice. After injection of CFA for 24 h, **CPIPC** (1, 3, 10 mg/kg), ibuprofen (2 mg/kg) or saline was orally administered, while mice in the control group were administered the same volume saline. The paw withdrawal mechanical threshold (PWMT) was measured at time points 0, 1, 2, 3 and 4 h after administration of drugs with a von Frey hair by an up-and-down method [35,36]. Mice were individually placed into transparent boxes with a metal mesh for about 1 h. Then, a series of von Frey hairs with different forces (0.4 g as a starter before 0.008, 0.02, 0.04, 0.07, 0.16, 0.60, 1.0, 1.4 and 2.0 g) were applied to stimulate the plantar of hind paws until a bend in a slight S-shape was observed for 5 to 6 s. If a paw withdrawal reaction was observed, the force was marked as positive; otherwise, it was marked as negative. Once crossover was found, the test was continued four times. The PWMT was calculated by a formula: PWMT (g) = 10^[Xf+Kδ]^, where Xf is the log value of the last von Frey hair force, K is the value retrieved from the standardized table based on the up-and-down pattern.

### 4.10. Formalin–Induced Inflammatory Pain Model

Adult male C57BL/6 mice were placed individually into transparent cages for about 1 h and then intraplantarally injected 20 µL formalin (5% formalin in vehicle saline) into the plantar surface of right hind paw. The duration of licking behaviors was recorded as a nociceptive response. The licking time was divided into two phases: phase I (0–5 min) and phase II (15–45 min). In the control group, mice were administered saline (0.1 mL/10 g). In the positive control group, ibuprofen (2 mg/kg) was orally administrated into mice for 1 h before formalin injection. In experimental groups, **CPIPC** (1, 3, 10 mg/kg) was intraperitoneally injected into mice for 1 h before formalin injection.

### 4.11. Measurement of Eye-Wiping

The pungency of capsaicin and CPIPC was evaluated in eye-wiping assay, using a protocol described previously. 10 μL of capsaicin (10 μg/mL) or **CPICP** (10, 30, 100 μg/mL) was dropped into mouse left eyes. In the vehicle group, the same volume of solvent was dropped into the left eyes of mice. The number of eye wipes was recorded after chemical administration for 60 s.

### 4.12. Randomization and Blinding

In vivo studies, animals were randomly distributed in the different experimental groups with each group having the same number of animals. Investigators for outcome assessments were blinded to group allocation. No blinding or randomization methods were used for in vitro studies.

### 4.13. Molecular Docking

The crystal structure of TRPV1 in a complex with RTX (PDB:5IRX) was obtained from PDB bank (http://www.rcsb.org/, accessed on 10 January 2022). The existing ligand, water and ions were removed from the PDB file. Molecular docking was performed via a Molecular Operating Environment (MOE v201608). The binding pocket was composed of residues Tyr511, Ser512, Leu515, Thr550, Asn551, Leu553, Tyr554, Arg557, Ala566, Ile569, Ile573 and Leu577 between S4 and S4–S5 linker. The 20 best docking results were exported and analyzed. The exported ligand-TRPV1 complexes were edited using UCSF Chimera 1.15 software (RBVI, University of California, San Francisco, CA, USA).

### 4.14. Statistical Analysis

Statistical analysis was performed by GraphPad Prism 8. Each data point is expressed as the mean ± SEM. Dose–response curves were fitted with a hill equation: y = V_max_x^n^/(k^n^ + x^n^), where V_max_ was the maximum effect, x was the drug concentration, k was EC_50_ (half maximal effective concentration) and n was the hill coefficient of sigmoidicity. Statistical analysis for differences between groups was performed by paired Student’s *t*-tests. For multiple comparisons among the groups for antinociceptive experiments, data were analyzed using one-way or two-way ANOVA followed by Bonferroni post hoc tests. A value of *p* < 0.05 was considered to have statistical significance.

## Figures and Tables

**Figure 1 molecules-27-05428-f001:**
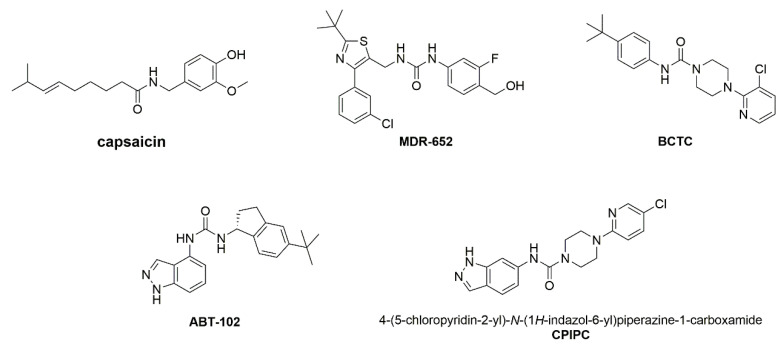
The chemical structures of TRPV1 modulators of capsaicin, MDR-652, BCTC, ABT-102 and **CPIPC**.

**Figure 2 molecules-27-05428-f002:**
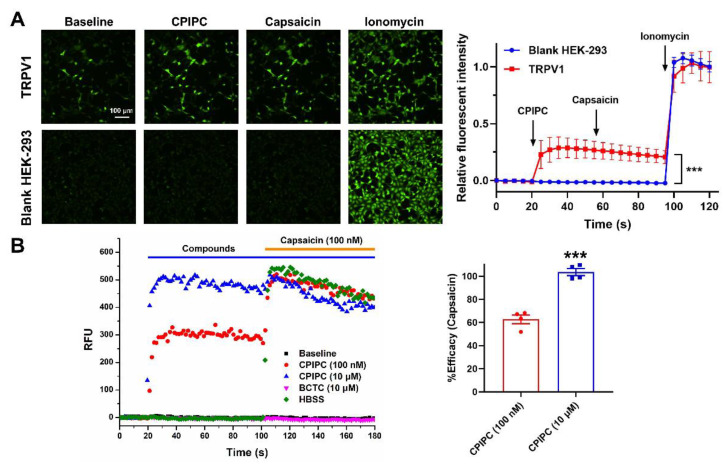
**CPIPC** induces Ca^2+^ influx in TRPV1 transiently transfected HEK293 cells. (**A**) Left: calcium-imaging of TRPV1 transiently transfected or blank HEK293 cells after treatment with **CPIPC** (1 μM), capsaicin (1 μM) and ionomycin (10 μM). Scale bar: 100 μm. Right: relative fluorescent intensity of Fluo-8 induced by **CPIPC**, capsaicin and ionomycin in TRPV1-expressing HEK293 cells (red square) and non-transfected cells (blue circle). The results of relative fluorescent intensity were calculated based on 45 cells in a glass coverslip (*n* = 45). Fluorescence intensity was normalized with the maximal value of ionomycin (10 μM). ***, *p* < 0.001. (**B**) Left: 100 nM (red) and 10 μM (blue) **CPIPC**-induced intracellular calcium rise and 10 μM BCTC (green, TRPV1 antagonist) blocked the activation effect of 10 μM **CPIPC** on TRPV1 channels in TRPV1-expressing HEK-293 cells in FlexStation 3 calcium fluorescence assay. Right: the efficacy of **CPIPC** (100 nM and 10 μM). Normalized by Capsaicin (100 nM) in FlexStation 3 calcium fluorescence assay (*n* = 4). ***, *p* < 0.001. RFU: Relative Fluorescence Unit.

**Figure 3 molecules-27-05428-f003:**
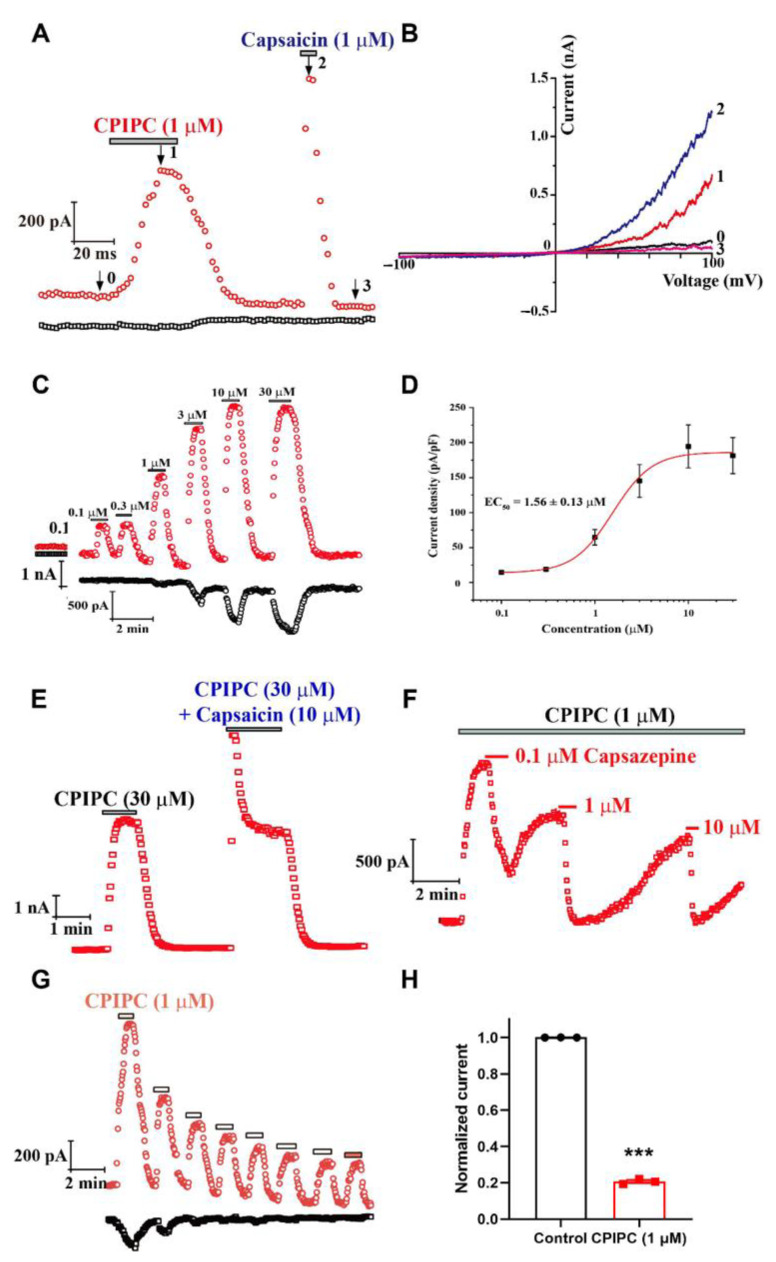
Concentration-dependent activation of TRPV1 by **CPIPC** in TRPV1 expressing HEK293 cells. (**A**) Time courses of TRPV1 current in response to 1 μM **CPIPC** and capsaicin measured at +100 mV (red) and −100 mV (black). (**B**) Representative whole-cell currents of TRPV1 in response to voltage ramps from −100 to +100 mV under control conditions (*0*), 1 μM **CPIPC** (*1*), 1 μM capsaicin (*2*), and wash-out (*3*) as indicated in (**A**). (**C**) Time courses of TRPV1 channels activated by different concentrations of **CPIPC** from 0.1 μM to 30 μM at +100 mV (red) and −100 mV (black). (**D**) Curve-fitting analysis of dose-dependent activation of TRPV1 by **CPIPC** with an EC_50_ of 1.56 ± 0.13 μM (*n* = 5). EC_50_: half maximal effective concentration. (**E**) Time course of TRPV1 current in response to 30 μM **CPIPC** and 10 μM capsaicin recorded at +100 mV. (**F**) Time course of TRPV1 current in response to 1 μM **CPIPC** and TRPV1 antagonist capsazepine at 0.1, 1 and 10 μM. (**G**) Time courses of TRPV1 current in response to repetitive applications of 1 μM **CPIPC** recorded at +100 mV (red) and −100 mV (black) in TRPV1 transiently transfected HEK293 cells. (**H**) The summary of normalized TRPV1 current after eight repetitive applications of 1 μM **CPIPC** (*n* = 3). ***, *p* < 0.001, for comparison of desensitization between the first and eighth application of **CPIPC**-induced currents.

**Figure 4 molecules-27-05428-f004:**
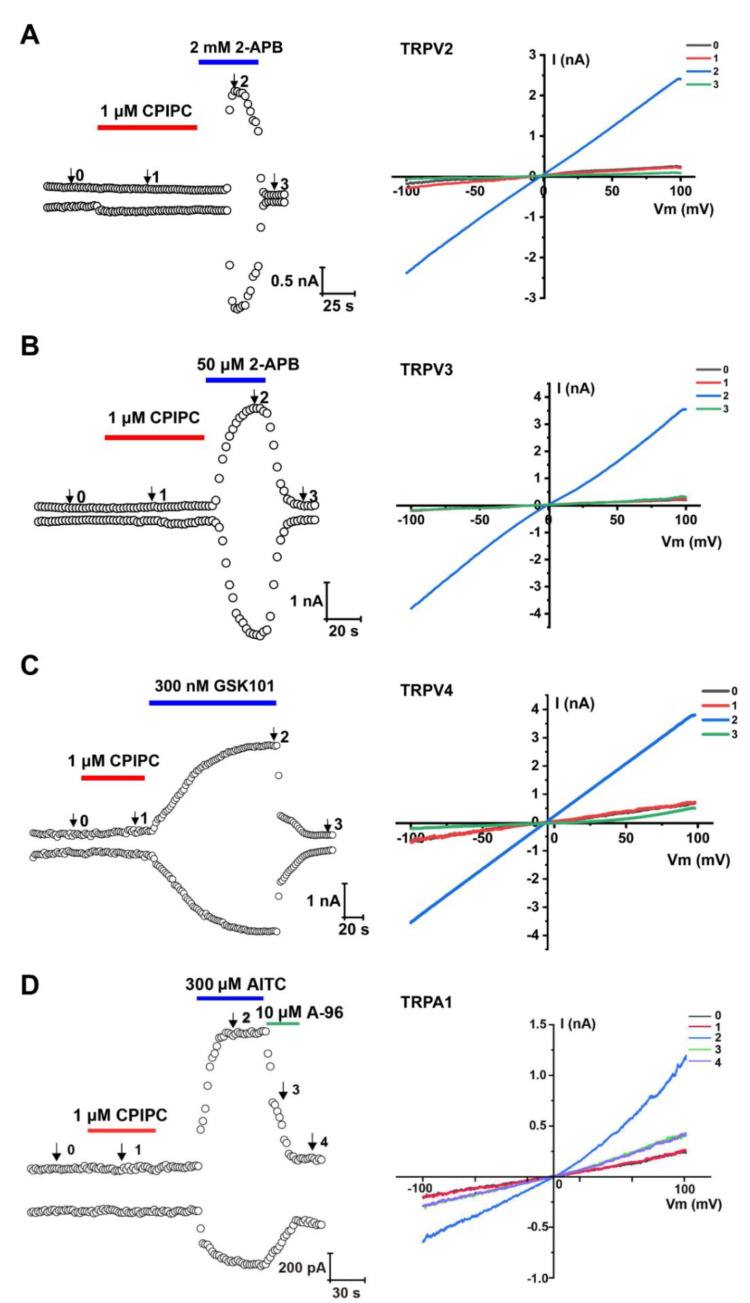
Lack of activation of *m*TRPV2, *h*TRPV3, *h*TRPV4 and *h*TRPA1 channels expressed in HEK293 cells by **CPIPC**. (**A**) Left: time courses of whole-cell *m*TRPV2 currents in response to 1 μM **CPIPC** and 2 mM 2-APB (TRPV2 agonist) measured at +100 mV (up) and −100 mV (down). Right: current-voltage curves of TRPV2 in response to voltage ramps from −100 to +100 mV under control conditions (*0*), addition of 1 μM of **CPIPC** (*1*), 2 mM of 2-APB (*2*) and washout (*3*). (**B**) Left: time courses of whole-cell *h*TRPV3 currents in response to 1 μM **CPIPC** and 50 μM 2-APB (TRPV3 agonist) measured at +100 mV (up) and −100 mV (down). Right: current-voltage curves of TRPV3 in response to voltage ramps from −100 to +100 mV under control conditions (*0*), addition of 1 μM of **CPIPC** (*1*), 50 μM of 2-APB (*2*) and washout (*3*). (**C**) Left: time courses of whole-cell *h*TRPV4 currents in response to 1 μM **CPIPC** and 300 nM GSK1016790A (TRPV4 agonist) measured at +100 mV (up) and −100 mV (down). Right: current-voltage curves of TRPV4 in response to voltage ramps from −100 to +100 mV under control conditions (*0*), addition of 1 μM of **CPIPC** (*1*), 300 nM GSK1016790A (*2*) and washout (*3*). (**D**) Left: time courses of whole-cell *h*TRPA1 currents in response to 1 μM **CPIPC**, 300 μM AITC (TRPA1 agonist) and 10 μM A967079 (TRPA1 antagonist) measured at +100 mV (up) and −100 mV (down). Right: current-voltage curves of TRPA1 in response to voltage ramps from −100 to +100 mV under control conditions (*0*), addition of 1 μM of **CPIPC** (*1*), AITC (*2*), A967079 (*3*) and washout (*4*).

**Figure 5 molecules-27-05428-f005:**
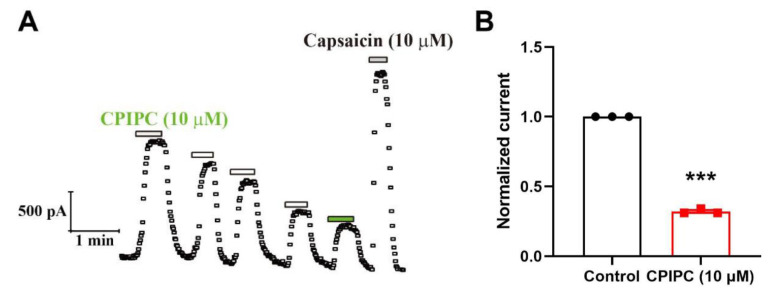
Induction of outward current desensitization by repetitive **CPIPC** in capsaicin-sensitive DRG neurons. (**A**) Time course of outward currents before and after application of 10 μM **CPIPC** and capsaicin measured at +80 mV. (**B**) Normalized outward current amplitude for comparison between first and last applications of 10 μM **CPIPC** (*n* = 3). ***, *p* < 0.001.

**Figure 6 molecules-27-05428-f006:**
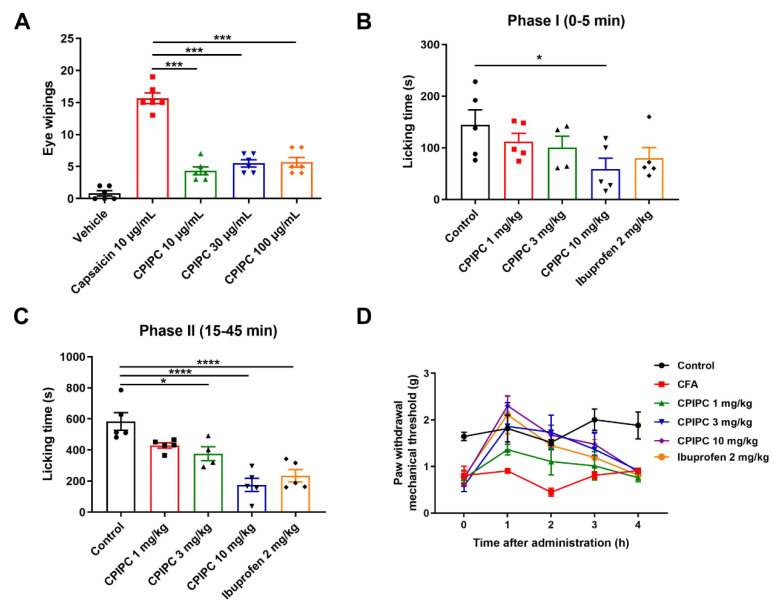
Pungency and antinociception of **CPIPC** in mice. (**A**) Effect of capsaicin (10 μg/mL) or CPIPC (10, 30 and 100 μg/mL) on the number of wiping movements evoked by instillation of 10 μL different solutions into mouse left eyes. Number of eye wipes was counted for 60 s after application of the compounds (*n* = 6). Statistical difference between the capsaicin and **CPIPC** groups is indicated as *** *p* < 0.001 by one-way ANOVA and *post hoc* Bonferroni’s test. (**B**,**C**) Anti-nociceptive effects of **CPIPC** on licking of inflammatory pain induced by intraplantar injection of formalin (5% formalin in vehicle saline) in the hind paw (*n* = 5). Phase I and II referring to the sum of licking time from 0 to 5 min and 15 to 45 min after the injection of formalin. Ibuprofen (2 mg/kg) and **CPIPC** (1, 3, 10 mg/kg) were pre-administered for 1 h before the intraplantar injection of formalin. Statistical difference between the vehicle control and administration groups is indicated as * *p* < 0.05, **** *p* < 0.0001 by one-way ANOVA and *post hoc* Bonferroni’s test. (**D**) Antinociceptive effects of **CPIPC** on inflammatory pain induced by intraplantar injection of 20 μL CFA (50% in saline) (*n* = 6). The paw withdrawal mechanical threshold (PWMT) of mice was measured after intraplantar injection of CFA for 24 h by von Frey hair. PWMT was measured at 0, 1, 2, 3, 4 h after oral administration of ibuprofen (2 mg/kg) and **CPIPC** (1, 3, 10 mg/kg). Statistic difference between the vehicle control and different doses of **CPIPC** groups is presented in Table 1.

**Figure 7 molecules-27-05428-f007:**
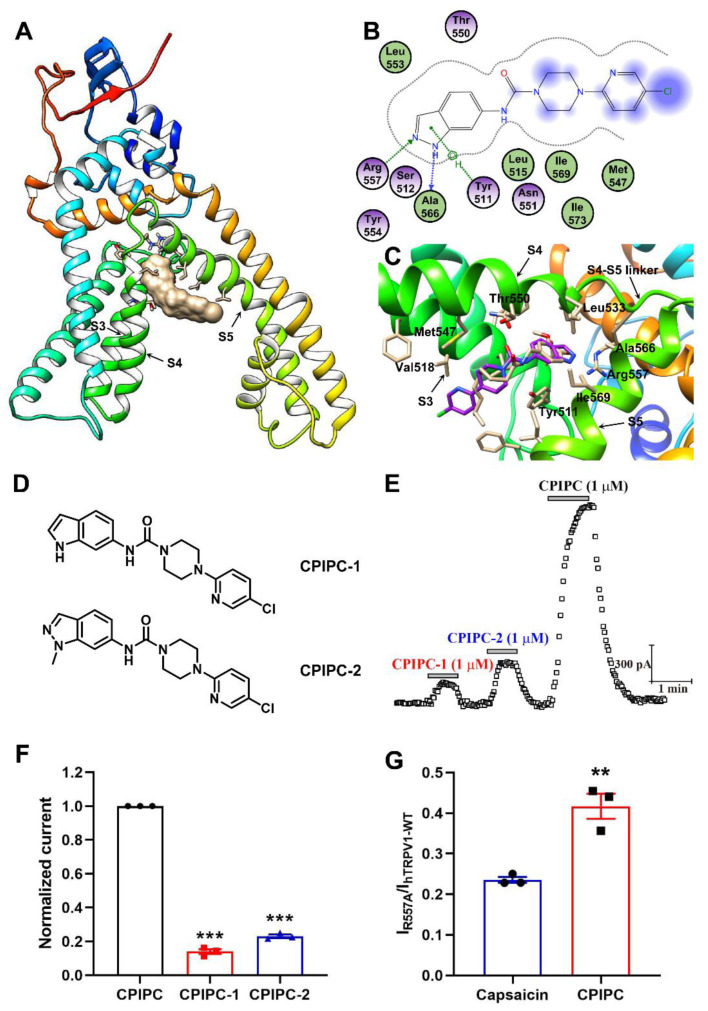
Identification of Arg557 as a key residue for opening of TRPV1 by **CPIPC**. (**A**) Location of ligand **CPIPC** and surrounding key residues in the crystal structure of *r*TRPV1 between S4 and S4–S5 linker (PDB: 5IRX). The surface of **CPIPC** was shown in gray, and the surrounding residues were shown in stick. (**B**) Molecular interactions between **CPIPC** and related residues of *r*TRPV1. Green and blue arrows: hydrogen bonds. (**C**) Overlap of **CPIPC** (purple) and capsaicin (cream) in the binding pocket of *r*TRPV1. (**D**) Chemical structure of **CPIPC** derivatives **CPIPC-1** and **CPIPC-2**. (**E**) Time course of TRPV1 currents in the presence and absence of **CPIPC** derivatives **CPIPC-1** (1 μM) and **CPIPC-2** (1 μM) recorded at +100 mV on *h*TRPV1-expressing HEK-293 cells. (**F**) The comparison of *h*TRPV1 current in response to **CPIPC**, **CPIPC-1** and **CPIPC-2** (1 μM) (*n* = 3). ***, *p* < 0.001. (**G**) The comparison of *h*TRPV1-WT and R557A mutant currents in response to capsaicin (1 μM) and **CPIPC** (1 μM) (*n* = 3). **, *p* < 0.01.

**Table 1 molecules-27-05428-t001:** Effects of **CPIPC** (*p.o.*) on mechanical paw withdrawal threshold parameters of inflammatory pain induced by CFA in mice.

Compounds (*p.o.* mg/kg)	Paw Withdrawal Mechanical Threshold (g)				
	0 h	1 h	2 h	3 h	4 h
Control	1.65 ± 0.09	1.82 ± 0.29	1.52 ± 0.06	2.00 ± 0.23	1.88 ± 0.29
CFA	0.80 ± 0.20 *	0.91 ± 0.02 *	0.45 ± 0.09 *	0.81 ± 0.07 **	0.91 ± 0.02 *
CPIPC (1)	0.75 ± 0.09	1.36 ± 0.11	1.11 ± 0.29	1.01 ± 0.30	0.76 ± 0.09
CPIPC (3)	0.58 ± 0.12	1.86 ± 0.51 ^#^	1.73 ± 0.37 ^##^	1.37 ± 0.37	0.91 ± 0.04
CPIPC (10)	0.74 ± 0.16	2.30 ± 0.21 ^##^	1.68 ± 0.21 ^##^	1.48 ± 0.24	0.88 ± 0.03
Ibuprofen (2)	0.78 ± 0.15	2.11 ± 0.41 ^##^	1.45 ± 0.08 ^#^	1.19 ± 0.39	0.82 ± 0.10

Data were expressed as the mean ± SEM (*n* = 6–8). Statistical difference is indicated as * *p* < 0.05, ** *p* < 0.01 vs. control; ^#^
*p* < 0.05, ^##^
*p* < 0.01 vs. CFA by two-way ANOVA and post hoc Bonferroni’s test.

## Data Availability

The data used to support the findings of this study are available. Further inquiries can be directed to the corresponding authors.

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
