# Peer review of "Identification of a Partial and Selective TRPV1 Agonist CPIPC for Alleviation of Inflammatory Pain"

_molecules, 2022, doi:10.3390/molecules27175428_

Round 1

Reviewer 1 Report

The study „Identification of a Partial and selective TRPV1 agonist CPIPC for Alleviation of Inflammatory Pain” by Dong et al. characterizes the newly synthesized partial agonist CPIPC. The manuscript is well written and the results are of high interest for further development of TRPV1 selective drugs. However, the manuscript would benefit from adressing some major points:

The authors focused on previously described interaction with R557. Indeed, this residue is highly important for the agonistic mechanism. Binding of agonists like capsaicin provoke a salt bridge between R557 and E570, that induces the movement of S4S5 linker towards S4 and subsequently opens the ion channel (Nature 2016;534:347–351). On the other hand, further interactions with rTRPV1 Y511 and S512 (Cell 2002;108:421–430) as well as M547 and T550 (J Biol Chem 2004;279:20283–20295) are needed to achieve an agonistic effect. Therefore, direct interactions alone with R557 and the compound do not directly lead to agonistic effects. The manuscript would benefit from comparison to the literature that is mentioned by Seebohm et al. 2021 (Cell Physiol Biochem 2021;55(S3):108-130).

Line 242-Line 246: The authors said, that mutating R557 resulted in a channel, which is insensitive to 1 µM CPIPC. On the other hand, Fig 7G as well as Figure S4B clearly show, that 1 µM CPIPC is able to elevate currents achieving around 40 % of current elevation observed at wildtype TRPV1. I suggest repeating the mutational analysis with a more suitable residue like Y511 or to rephrase the Lines 242-246. Mutation of Y511 to alanine leads to a complete loss of capsaicin activity.

Further, the authors should check the following minor points:

·       Fig 2A: scale bar label should be added to the figure instead of figure legend

·       Fig 2A/2B: please add statistics/p values for Figure right panels

·       Line 127: I suggest adding species abbreviations to the results part, since pharmacology of TRPV1-4 is highly species dependent.

·       I suggest to use identical color code for Fig 7A and Fig 7C. Furthermore, captions for secondary structures like S4 helix, S4S5 linker, etc would be helpful in both panels.

·       Fig 7B: font size of amino acids should be increased.

·       Please add statistics/p values to Figure 7G

·       Please add statistics/p values to Fig S3

·       Line 230: Ala566 instead of Ser566

·       Line 244: Figure S4 instead of Figure S3

Author Response

REVIEWER #1

The study, Identification of a Partial and selective TRPV1 agonist CPIPC for Alleviation of Inflammatory Pain” by Dong et al. characterizes the newly synthesized partial agonist CPIPC. The manuscript is well written and the results are of high interest for further development of TRPV1 selective drugs. However, the manuscript would benefit from adressing some major points:

Question #1: The authors focused on previously described interaction with R557. Indeed, this residue is highly important for the agonistic mechanism. Binding of agonists like capsaicin provoke a salt bridge between R557 and E570, that induces the movement of S4S5 linker towards S4 and subsequently opens the ion channel (Nature 2016;534:347–351). On the other hand, further interactions with rTRPV1 Y511 and S512 (Cell 2002;108:421–430) as well as M547 and T550 (J Biol Chem 2004;279:20283–20295) are needed to achieve an agonistic effect. Therefore, direct interactions alone with R557 and the compound do not directly lead to agonistic effects. The manuscript would benefit from comparison to the literature that is mentioned by Seebohm et al. 2021 (Cell Physiol Biochem 2021;55(S3):108-130).

Response #1: As suggested, we added the descriptions for comparing those key residues Y511, S512, M547 and T550 in the Discussion in this revision (Line 287-291) and also cited these references as references 39-41. Our molecular docking reveals that CPIPC and capsaicin share a similar pose in the binding pocket formed by S4 and S4-S5 linker. Either indazole of CPIPC or phenolic hydroxyl group of capsaicin can form a hydrogen bond with Arg557, whereas mutating R557 alters CPIPC- or capsaicin-induced channel sensitivity. Besides R557, residues Y511 and S512 from S3 segment and M547 and T550 from S4 segment are also essential for agonistic effect of capsaicin, as mutations Y511A or S512Y of rTRPV1 abolish the channel sensitivity to capsaicin, indicating that these key residues also contribute to CPIPC-mediated activation of TRPV1.

Question #2: Line 242-Line 246: The authors said, that mutating R557 resulted in a channel, which is insensitive to 1 µM CPIPC. On the other hand, Fig 7G as well as Figure S4B clearly show, that 1 µM CPIPC is able to elevate currents achieving around 40 % of current elevation observed at wildtype TRPV1. I suggest repeating the mutational analysis with a more suitable residue like Y511 or to rephrase the Lines 242-246. Mutation of Y511 to alanine leads to a complete loss of capsaicin activity.

Response #2: As suggested, we rephrased the sentence by stating “Mutating Arg557 caused a dramatic attenuation of CPIPC- (or capsaicin-mediated TRPV1 activation (Figure 7G and Figure S4), which is consistent with the observation that residue Arg557 is critical for capsaicin-evoked response” in this revision (line 246-248).

Question #3: Further, the authors should check the following minor points:

Fig 2A: scale bar label should be added to the figure instead of figure legend

Response #3: As suggested, we added the scale bar label to the Figure 2A.

Question #4: Fig 2A/2B: please add statistics/p values for Figure right panels

Response #4: As suggested, we added the p values in the right panel of Figure 2A/2B.

Question #5: Line 127: I suggest adding species abbreviations to the results part, since pharmacology of TRPV1-4 is highly species dependent.

Response #5: As suggested, we added the species abbreviations of mTRPV2, hTRPV3, hTRPV4, hTRPA1 and hNav1.7 channels in this revision (Line 129-130).

Question #6: I suggest to use identical color code for Fig 7A and Fig 7C. Furthermore, captions for secondary structures like S4 helix, S4S5 linker, etc would be helpful in both panels.

Response #6: As suggested, an identical color code was used for Figure 7A and 7C.

Question #7: Fig 7B: font size of amino acids should be increased.

Response #7: The font size was increased for Figure 7B.

Question #8: Please add statistics/p values to Figure 7G

Response #8: As suggested, we added the p values for Figure 7G.

Question #9: Please add statistics/p values to Fig S3

Response #9: As suggested, we added the p values for Figure S3.

Question #10: Line 230: Ala566 instead of Ser566

Response #10: As suggested, we corrected wiht “Ala566” in this revision (line 233).

Question #11: Line 244: Figure S4 instead of Figure S3

Response #11: As suggested, we corrected it in this revision.

Reviewer 2 Report

The authors have created a new TRPV1 agonist, CPIPC, and found that CPIPC provided antinociceptive effects with mice. Overall, the findings are fantastic in this pain research field, but in some areas in this manuscript, I request that the authors should correct some mistakes and re-write some texts.

In some sentences in the manuscript, the authors should add a space between words. For example, I have found ‘Leu670[22]’ in L54 and ‘1.6 s[12,25,32]’ in L347. In other areas, I have found some typos. For example, ‘coverslip, (n = 45)’ in L92 should be replaced with ‘coverslip (n = 45)’. Moreover, ‘depolarizating’ in L143 should be ‘depolarizing’, I strongly suggest that the authors should peruse, not skim, the whole manuscript and/or should have the manuscript English-proofread by some services.

For Fig 2A, why is the TRPV1-transfected HEK293 cells at the baseline bright? Is there any regular Ca influx at the baseline?

Regarding Fig 2, add some description about the role of ionomycin to the main text. In addition, what is RFU in Fig 2B? I could not find the definition of RFU or how RFU was calculated in the main text.

For Fig 3A, C, E, G, why did the authors provide the current (pA) traces in a consecutive circle plot? Normally, the current/voltage traces are drawn in a line plot. Are these figures not traces?

For 3D, the authors should describe how to calculate the current density in some areas in the main text. If you have already done, where is the description?

For Fig 3E and 3F, the traces should be drawn in red, maybe? That improves readability.

In L139, why was the voltage held at +80 mV?

In L140, ‘53% was sensitive’, so does the remaining 47% have any roles?

For Fig 4 A-D (left), at what voltage (mV) did the membrane held?

In L167, the manuscript says, ‘measured at +100 mV’, but also says, ‘DRG neurons held at +80 mV’ in L139. Are these consistent?

In Discussion, the authors have discussed structural properties of CPIPC and TRPV1, and pharmacological potentials of CPIPC. To make this paper more attractive, do you have any consideration about the behavioral effect of CPIPC on mice (ie, Fig 6)? For example, (i) what CFA is, (ii) the relationship between paw withdrawal and time after treatment (Fig 6D), (iii) the difference between CPIPC and other TRPV1 agonists/antagonists, (iv) the difference between Fig 6B and 6C, and so on. Anything will be welcome.

Again, the methods and figures should be more concrete and descriptive; eg, how the authors calculated parameters and so on.

Author Response

REVIEWER #2

The authors have created a new TRPV1 agonist, CPIPC, and found that CPIPC provided antinociceptive effects with mice. Overall, the findings are fantastic in this pain research field, but in some areas in this manuscript, I request that the authors should correct some mistakes and re-write some texts.

Question #1: In some sentences in the manuscript, the authors should add a space between words. For example, I have found ‘Leu670[22]’ in L54 and ‘1.6 s[12,25,32]’ in L347. In other areas, I have found some typos. For example, ‘coverslip, (n = 45)’ in L92 should be replaced with ‘coverslip (n = 45)’. Moreover, ‘depolarizating’ in L143 should be ‘depolarizing’, I strongly suggest that the authors should peruse, not skim, the whole manuscript and/or should have the manuscript English-proofread by some services.

Response #1: As suggested, we added some space between words in the manuscript, and revised “coverslip, (n = 45), depolarizating” into “coverslip (n = 45), depolarizing”. In addition, we perused and corrected those errors previously appeared in the manuscript.

Question #2: For Fig 2A, why is the TRPV1-transfected HEK293 cells at the baseline bright? Is there any regular Ca influx at the baseline?

Response #2: The green fluorescent proteins as tags were also expressed in TRPV1-transfected HEK293 cells, therefore, TRPV1-transfected HEK293 cells are brighter than non-transfected cells at the baseline.

Question #3: Regarding Fig 2, add some description about the role of ionomycin to the main text. In addition, what is RFU in Fig 2B? I could not find the definition of RFU or how RFU was calculated in the main text.

Response #3: As suggested, we added the description for the role of ionomycin as a control in the text by stating “Ionomycin, as a positive control of calcium ionophore, resulted in a drastic elevation of intracellular calcium in both TRPV1 overexpressing and non-transfected HEK293 cells.” (Line 82-84, in the revision).

The RFU is the acronym of Relative Fluorescence Unit that is readout from the assay of FlexStation 3 multi-mode microplate reader. We added this information in the legend of Figure 2 (Line 98 in this revision).

Question #4: For Fig 3A, C, E, G, why did the authors provide the current (pA) traces in a consecutive circle plot? Normally, the current/voltage traces are drawn in a line plot. Are these figures not traces?

Response #4: For Figure 3A, C, E, G, current amplitudes were measured and plotted as circles at different time points. Those figures are presented as not continuous traces.

Question #5: For 3D, the authors should describe how to calculate the current density in some areas in the main text. If you have already done, where is the description?

Response #5: As suggested, we added the description for calculation of current density as “Divide the whole-cell current (Im) by the cell capacitance (Cm) to calculate the current density (Id).” (Line 385-386 in this revision).

Question #6: For Fig 3E and 3F, the traces should be drawn in red, maybe? That improves readability.

Response #6: As suggested, the traces in red were drawn for Figure 3E and 3F.

Question #7: In L139, why was the voltage held at +80 mV?

Response #7: As described in previous reports, a testing pulse for DRG neurons was applied at +80 mV [1, 2]. Therefore, we tested the effect of CPIPC on DRG neurons at +80 mV.

Question #8: In L140, ‘53% was sensitive’, so does the remaining 47% have any roles?

Response #8: Not all small- and medium-diameter nociceptive neurons in DRG were capsaicin-sensitive [1, 3]. In this study, 53% of DRG neurons responded to capsaicin with outward currents, whereas the remaining 47% was insensitive to capsaicin likely due to lack of or low expression of TRPV1.

Question #9: For Fig 4 A-D (left), at what voltage (mV) did the membrane held?

Response #9: The membrane was held at -100 mV (up) and +100 mV (down). As suggested, we added the description in the Figure 4 legend (Line 152-168 in this revision).

Question #10: In L167, the manuscript says, ‘measured at +100 mV’, but also says, ‘DRG neurons held at +80 mV’ in L139. Are these consistent?

Response #10: Yes, we re-checked that we tested the effect of CPIPC on TPRV1-4 in HEK293 cells held at +100 mV and DRG neurons held at +80 mV.

Question #11: In Discussion, the authors have discussed structural properties of CPIPC and TRPV1, and pharmacological potentials of CPIPC. To make this paper more attractive, do you have any consideration about the behavioral effect of CPIPC on mice (ie, Fig 6)? For example, (i) what CFA is, (ii) the relationship between paw withdrawal and time after treatment (Fig 6D), (iii) the difference between CPIPC and other TRPV1 agonists/antagonists, (iv) the difference between Fig 6B and 6C, and so on. Anything will be welcome.

Response #11: We added the description in the Discussion (Line 260-271).

(i) Complete Freund’s Adjuvant (CFA) is a widely used reagent for generation of rodent models of inflammation.

(ii) The injection of CFA into the hindpaws of mice or rats results in both mechanical allodynia and thermal hyperalgesia within 24 h. The inflammation condition occurs for 5 days before a slow recovery [35,36].

(iii) CPIPC (1, 3, 10 mg/kg) dose-dependently alleviated the CFA-induced mechanical allodynia and the effect lasted for about 2 h in mice. CPIPC as a partial TRPV1 agonist is different than other TRPV1 agonist such capsaicin that causes burning sensation and is only for topical use. CPIPC can be internally used, holding developmental potential for antinociceptive agent.

(iv) Injection of formalin into mouse hindpaws induces pain response in two phases: phase I (Figure 6B), an acute pain stage caused by direct stimulation of C-fiber nociceptors; and phase II (Figure 6C), as later phase of inflammatory response [32].

Question #12: Again, the methods and figures should be more concrete and descriptive; eg, how the authors calculated parameters and so on.

Response 12: As suggested, we perused and added more descriptions for methods and figures in this revision (Line 448-455). Statistics was analyzed using GraphPad Prism 8 program. Each data point is expressed as the mean ± SEM. Dose-response curves were fitted with a hill equation: y = Vmaxxn/(kn + xn), where Vmax was the maximum effect, x was the drug concentration, k was EC50 (half maximal effective concentration) and n was the hill coefficient of sigmoidicity. Statistical analysis for differences between groups was performed by paired Student’s t tests. For multiple comparisons among the groups for antinociceptive experiments, data were analyzed using one-way or two-way ANOVA followed by Bonferroni post hoc tests. A value of p < 0.05 was considered to be statistical significance.

References cited in this letter

  1. Wei, N. N.; Lv, H. N.; Wu, Y.; Yang, S. L.; Sun, X. Y.; Lai, R.; Jiang, Y.; Wang, K., Selective Activation of Nociceptor TRPV1 Channel and Reversal of Inflammatory Pain in Mice by a Novel Coumarin Derivative Muralatin L from Murraya alata. J Biol Chem 2016, 291, (2), 640-51.
  2. Yang, S.; Yang, F.; Wei, N.; Hong, J.; Li, B.; Luo, L.; Rong, M.; Yarov-Yarovoy, V.; Zheng, J.; Wang, K.; Lai, R., A pain-inducing centipede toxin targets the heat activation machinery of nociceptor TRPV1. Nat Commun 2015, 6, 8297.
  3. Su, X.; Wachtel, R. E.; Gebhart, G. F., Capsaicin sensitivity and voltage-gated sodium currents in colon sensory neurons from rat dorsal root ganglia. The American journal of physiology 1999, 277, (6), G1180-8.

Round 2

Reviewer 1 Report

The authors revised their manuscript "Identification of a partial and selective TRPV1 agonist CPIPC for alleviation of inflammatory pain" and clearly improved the presentation of the results. All aspects that I mentioned were adressed. I would suggest the manuscript for publication in the present form.